

**Critical Evaluation of Strong Ground Motions in Izmir and Implications for Future Earthquake Simulation**
**Results**
Sahin Caglar Tuna
Ass. Prof. Dr.; Yasar University, Izmir
**ABSTRACT**
Izmir, a major city in western Turkey, is located in a highly seismic region, subject to frequent earthquakes due to
its proximity to active fault systems. This paper critically evaluates the strong ground motions recorded in Izmir,
with a focus on understanding the implications for urban infrastructure and future seismic hazard mitigation.
Historically available data is collected and compared with the available ground motion prediction equations
(GMPE). Later, the most appropriate prediction equation is selected and used to determine the target response
spectrum. 2020 Sisam earthquake is a well-documented seismic event and the data from the stations are then used
to further calibrate the 1D site response model. Lastly, possible future events are generated and results are
compared with the current Turkish Earthquake Code (TEC). Amplification factors prescribed by code for İzmir
Bay have been surpassed by projected future events, highlighting the necessity for reassessment. Therefore, region-
specific seismic zoning should be established when standard code practices fall short in accounting for significant
site effects. Concrete recommendations about local site modification factors and evaluations on this topic have
been provided within the article.
Keywords: Ground motion prediction equations, Site response, Future events, Local site modification factors











## 1. INTRODUCTION

### 1.1. Scope and Aim

Izmir, Turkey's third-largest city, is located on the Aegean coast, and its proximity to active fault lines makes it highly vulnerable to seismic activity. Izmir is located within the extensional tectonic regime of the Aegean region, where several active faults, including the Izmir Fault and the Seferihisar Fault, contribute to the area's high seismic risk (Emre et al., 2018). The proximity of Izmir to the Hellenic subduction zone also increases its seismic hazard, as this plate boundary is responsible for generating frequent and potentially large earthquakes (McKenzie, 1978). In particular, shallow crustal earthquakes have historically caused significant ground shaking and damage in the region (Emre et al., 2005). Some of the recent studies have detailed the active faults in the region from which, the activity of the seismic hazard can be evaluated easily (Figure 1).

The city has been impacted by numerous destructive earthquakes throughout history. The 1688 and 1778 earthquakes were particularly devastating, with reports of widespread destruction (Tepe et.al. 2021). In more recent times, the 2020 Samos earthquake have provided critical data on the ground motions experienced in the region. These events highlighted the varying response of different local soil conditions and the importance of considering site-specific factors in seismic hazard assessment (Cetin et.al, 2022).

Given the city's dense population and economic importance, a critical evaluation of the ground motion characteristics during earthquakes is essential for improving preparedness and urban resilience. Buildings with poor design or inadequate retrofitting were particularly vulnerable, as they were not able to withstand the amplified seismic waves. Understanding these interactions is key to developing more effective risk mitigation strategies and informing future urban planning.

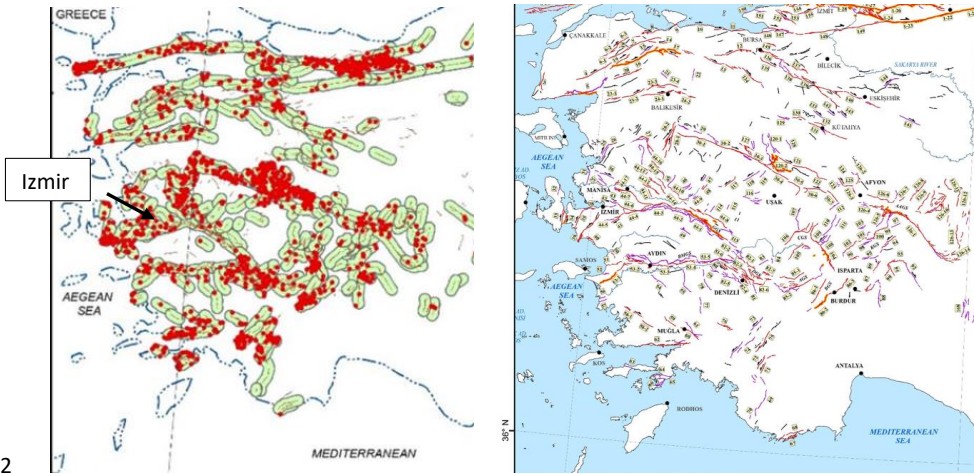

Figure 1. Active Seismic Faults and recent earthquakes in the region (Emre et.al, 2018)

The purpose of this study is divided into two main parts: First part is to evaluate the strong ground motions recorded in Izmir during past seismic events, particularly focusing on their effects on local geotechnical conditions and built



environments. In the second part, a future earthquake scenario and potential engineering outcomes will be
examined by using the findings obtained in the first part.
The steps involved in this study include:
a. Data Collection: Gathering historical earthquake data from the Izmir region, including earthquake
magnitudes, source-to-site distances, and PGA measurements.
b. Selection of GMPEs: Choosing GMPE models that are applicable to the regional tectonic and geological
conditions.
c. Comparison of GMPE Predictions and evaluation of GMPE Accuracy: Comparing the predicted PGA
values from different GMPEs with the observed values from historical earthquakes. Differences were
observed between the predicted and actual ground motions, emphasizing the importance of site-specific
adjustments in GMPEs for accurate seismic hazard assessment. Using statistical methods, such as Root
Mean Square Error (RMSE), to assess the accuracy of the GMPE predictions and identify the most
reliable model for the Izmir region. Apply necessary improvements for the prediction equations to comply
with the specific directivity and near fault effects.
d. 1D site response analysis were firstly validated with the available recordings and then set up for the future
earthquake scenarios.
e. Developing target spectra using the outcomes of 3rd step, evaluating future earthquakes in the region and
comparison with the current TEC results.
f. The study concludes with recommendations on refining seismic hazard models to account for local site
effects and improving the predictive accuracy of GMPEs in areas with complex soil profiles. These
findings have implications for earthquake-resistant design and site-specific seismic risk mitigation
strategies.
1.2. The Geological and Geotechnical Settings of Izmir Bay
The geological structure of Izmir is highly variable, consisting of sedimentary basins with alluvial soils and rock
outcrops. These heterogeneous ground conditions play a crucial role in amplifying seismic waves and influencing
the distribution of damage during earthquakes. This is particularly important in areas with soft soils or complex
geological features, which can greatly affect the intensity and frequency content of seismic waves at the surface.
As part of the Aegean region, Izmir is situated within an active tectonic zone characterized by extensional
processes and numerous fault systems, contributing to its significant seismic hazard. The Izmir Bay region is
located in the western part of Turkey and is part of the larger Aegean Extensional Province. This region is
influenced by the ongoing tectonic extension between the African and Eurasian plates, creating a highly active
fault system that includes both normal and strike-slip faults (Akyol et.al. 2006). The geological makeup of Izmir
Bay consists of a variety of rock types and sedimentary deposits that influence the behavior of seismic waves
during an earthquake:



- Sedimentary Basins: The region includes several sedimentary basins, including the Gediz Graben and the Menderes Massif. These basins are filled with younger, unconsolidated sediments that can amplify seismic waves.
- Alluvial Deposits: Much of the coastal region, including areas surrounding the bay, is composed of alluvial deposits. These sediments, deposited by rivers, are loosely consolidated and can exacerbate ground shaking during an earthquake.

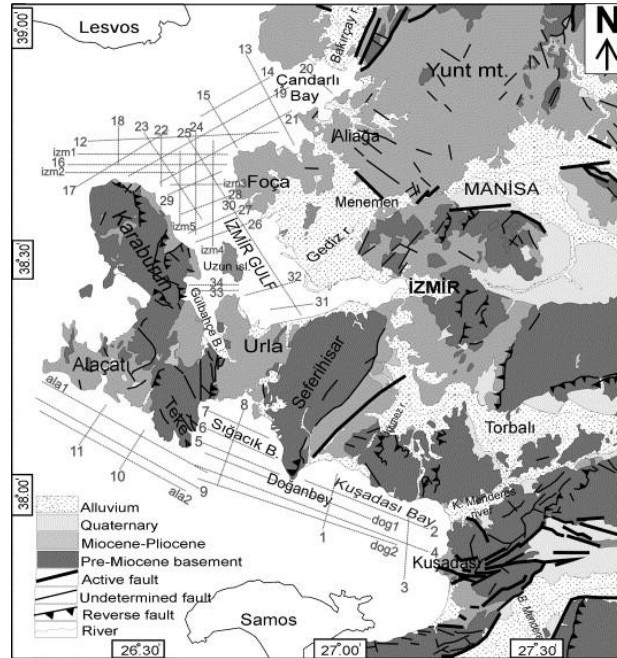

Figure 2. Geology map of the study area and location of the seismic fault lines (Adapted from Ocakoglu et.al, 2005).

## 2. Compilation of the strong motion dataset and predictive performance of current ground motion models

Ground Motion Prediction Equations (GMPEs) are empirical or semi-empirical mathematical models used to estimate the expected level of ground shaking (ground motion) at a specific location during an earthquake. GMPEs play a critical role in seismic hazard analysis and earthquake engineering by predicting key seismic parameters based on several factors such as earthquake magnitude, distance to the fault, and local site conditions (Gulerce et.al 2022). The Izmir region, located in Western Anatolia, is seismically active and has complex fault systems and varying soil conditions. Therefore, selecting an appropriate target spectrum for this region requires a detailed comparison of GMPE predictions with observed earthquake records.



For that aim, historical earthquake data from the Izmir region, including earthquake magnitudes, source-to-site
distances, and PGA measurements were gathered. Historical ground motion records were compiled from Turkish
Ministry of Interior Disaster and Emergency Management Presidency (AFAD). A total 33 earthquake events,
dating from 1996 to 2024 were selected and given in Table 1 with the recorded peak ground acceleration (PGA)
values. A total of 8 different GMPEs were used for comparison and validation purpose (Table 2). The predicted
peak ground acceleration (PGA) values from various GMPEs with the actual observed values from historical
earthquakes were compared (Figure 3).
Table 1.Important characteristics of the historical seismic events

| Event No | Event Name | Mw | Epicentral Distance - km | Fault Mechanism | Event Depth -km | PGA Max - cm/s2 |
|---|---|---|---|---|---|---|
| 1 | 10.04.2003 - Seferihisar | 5.7 | 37.45 | Strike Slip | 18.7 | 78.57 |
| 2 | 17.10.2005 - Urla | 5.8 | 58.21 | Strike Slip | 11 | 13.12 |
| 3 | 17.10.2005 - Urla | 5.4 | 56.17 | Strike Slip | 20.5 | 16.51 |
| 4 | 20.10.2005 - Urla | 5.9 | 58.98 | Strike Slip | 15.4 | 31.773 |
| 5 | 30.10.2022- Sisam | 7 | 75.57 | Normal | 16.54 | 73.72 |
| 6 | 12.06.2017-Karaburun | 6.2 | 43.87 | Normal | 15.86 | 58.306 |
| 7 | 11.04.2022- Buca - İzmir | 4.9 | 9.81 | Strike Slip | 14.47 | 48.59 |
| 8 | 19.07.2014 - Konak- İzmir | 3.7 | 10 | Strike Slip | 6.98 | 9.47 |
| 9 | 21.04.2021-Sehzadeler-Manisa | 4.9 | 40.19 | Normal | 13.2 | 9.673 |
| 10 | 12.06.2017-Karaburun | 6.2 | 89.62 | Normal | 15.86 | 25.499 |
| 11 | 26.06.2020- Saruhanlı-Manisa | 5.5 | 64.1 | Normal | 9.29 | 7.109 |
| 12 | 8.01.2013 - Aegean Sea | 6.2 | 194 | | 26.83 | 3.642 |
| 13 | 24.05.2014 - Aegean Sea | 6.5 | 255.78 | | 25 | 7.659 |
| 14 | 02.06.2017 Ayvacik (Canakkale) | 5.3 | 154.13 | Normal | 14.16 | 1.466 |
| 15 | 06.17.2017 Aegean Sea | 5.3 | 81.15 | Normal | 9.11 | 9.42 |
| 16 | 07.20.2017 Aegean Sea(Bodrum) | 6.5 | 169.93 | Normal | 19.44 | 4.44 |
| 17 | 18.02.2020 Kırkağaç (Manisa) | 5.2 | 91.02 | Normal | 6.98 | 4.662 |
| 18 | 19.05.2011 Simav (Kutahya) | 5.7 | 180.7 | Normal | 24.46 | 5.533 |
| 19 | 08.08.2019 Bozkurt (Denizli) | 6 | 218.49 | Normal | 10.92 | 0.39 |
| 20 | 28.06.2020 Ege Denizi (Mugla) | 5.2 | 214.57 | Normal | 61.42 | 3.375 |
| 21 | 30.10.2020 Sisam | 5.1 | 72.95 | Normal | 7.71 | 7.056 |
| 22 | 21.06.2021 Aegean Sea (Datca) | 5.3 | 228.34 | Normal | 14.74 | 0.61 |
| 23 | 01.10.2023 Lesvos | 5 | 130.95 | Normal | 14.95 | 1.708 |
| 24 | 27.01.2024 Aegean Sea (Kusadası) | 5.1 | 52.46 | Normal | 8.51 | 7.632 |
| 25 | 2.04.1996 | 4.9 | 71.81 | Normal | 12 | 18.42 |
| 26 | 14.11.1997 Aegean Sea (Kusadası) | 5.8 | 128.79 | Normal | 12 | 6.03 |
| 27 | 09.07.1998 Aegean Sea | 5 | 75.48 | Normal | 12.5 | 27.06 |
| 28 | 17.08.1999 Golcuk (Izmit) | 7.6 | 346.5 | Strike Slip | 15.9 | 10.8 |
| 29 | 21.01.2002 Turgutlu (Manisa) | 4.8 | 60.11 | Normal | 11.7 | 6.981 |
| 30 | 17.04.2003 Seferihisar (Izmir) | 5.2 | 61.55 | Strike Slip | 11.5 | 8.851 |
| 31 | 29.01.2005 | 4.9 | 47.67 | Normal | 20 | 6.131 |
| 32 | 22.01.1999 Buca-İzmir | 3.4 | 9.95 | Strike Slip | 5 | 2.985 |
| 33 | 24.12.2005 Akhisar (Manisa) | 4.9 | 64.85 | Normal | 6 | 3.14 |









Table 2. GMPEs Used In this Study

| NO | GROUND MOTION PREDICTION EQUATIONS |
|----|-------------------------------------|
| 1 | AS08: Abrahamson & Silva 2008 NGA Model |
| 2 | BA08: Boore & Atkinson 2008 NGA Model |
| 3 | CB08: Campbell & Bozorgnia 2008 NGA Model |
| 4 | CY08: Chiou & Youngs 2008 NGA Model |
| 5 | Abrahamson & Silva & Kamai 2014 NGA West-2 Model |
| 6 | Boore & Stewart & Seyhan & Atkinson 2014 NGA West-2 Model |
| 7 | Campbell & Bozorgnia 2014 NGA West-2 Model |
| 8 | Chiou & Youngs 2014 NGA West-2 Model |


To quantify the accuracy of GMPE predictions, error analysis were conducted using statistical metrics in which
the goal is to determine which GMPE provides the closest predictions to the observed data across various
earthquake magnitudes and site conditions. As per the error analysis in GMPE evaluations, two methods were
chosen, R^2 (Coefficient of Determination) and RMSE (Root Mean Square Error) and the results were given in
Table 3.
RMSE is a commonly used metric for quantifying the difference between observed and predicted values. It
measures the square root of the average of the squared differences between the predicted PGA values from GMPEs
and the actual observed values. RMSE gives more weight to larger errors, making it particularly useful when larger
deviations in predictions need to be minimized. RMSE can be calculated as:
$$\text{RMSE} = \sqrt{\frac{1}{N} \sum_{i=1}^{N} (PGA_{observed,i} - PGA_{predicted,i})^2}$$
Where:
-    N is the number of the earthquake records
-    PGA,observed,i is the observed PGA for the i-th earthquake
-    PGA, predicted,i is the predicted PGA for the i-th earthquake based on the GMPE.










Figure 3. The results of the analysis are given in the table below.



Table 3. Result of GMPE Error Analysis

| Root Mean Square Error | | | | | | | |
|---|---|---|---|---|---|---|---|
| RMSE / AS08 | RMSE / BA08 | RMSE / CB08 | RMSE / CY08 | RMSE / ASK14 | RMSE / BSSA14 | RMSE / CB14 | RMSE / CY14 |
| 14.85 | 13.53 | 17.11 | 16.75 | 15.95 | 13.39 | 11.02 | 14.51 |
| R^2 (Coefficient of Determination) | | | | | | | |
| R2 / AS08 | R2 / BA08 | R2 / CB08 | R2 / CY08 | R2 / ASK14 | R2 / BSSA14 | R2 / CB14 | R2 / CY14 |
| 0.50 | 0.59 | 0.55 | 0.46 | 0.55 | 0.80 | 0.71 | 0.56 |


There can be several factors for the observed differences between the models, for instance site and soil
amplification effects or the inconsistency in depth, magnitude or distance scaling of the models and the several
constants implemented in each models. As confinement of these effects and limiting the sampling data is not
possible in this study, typically the error analysis is compared and the resulting ranking is used for selecting the
two most powerful predictive equation. The results indicate that Model CB14 and BSSA14 are better choices for
the following analysis.
**3.    Site response validation analysis for future predicted events**
The next step for the generation of future earthquakes is to evaluate and correctly determine the site properties.
For that aim, 1D site response analysis (SRA) was set up and validated with the available records from 2020 Sisam
earthquake. For SRA's, Deepsoil software (Hashash et.al. 2020) was used as the program was previously used by
many researchers and the adaptive nature of the program was well calibrated (Cetin et.al. 2022).
For calibration and validation purpose, a well recorded and data riched event was needed. The 2020 Izmir
earthquake struck on October 30, 2020, with a moment magnitude (Mw) of 6.9. Its epicenter was located in the
Aegean Sea, approximately 14 kilometers northeast of the Greek island of Samos, but it caused significant damage
in Izmir due to its shallow depth and local site effects. Event was recorded by several seismograms located around
the city (Figure 4), some of which are located on alluvial plains and some on rock outcrops, which allows
researchers to further evaluate site effects (Kramer,1996). The city covers large areas of alluvial soil conditions,
therefore different regions were selected for the validation purposes. One of the site is located in Karsiyaka,
western part; the other sites are located in Konak- Bayrakli and Bornova.






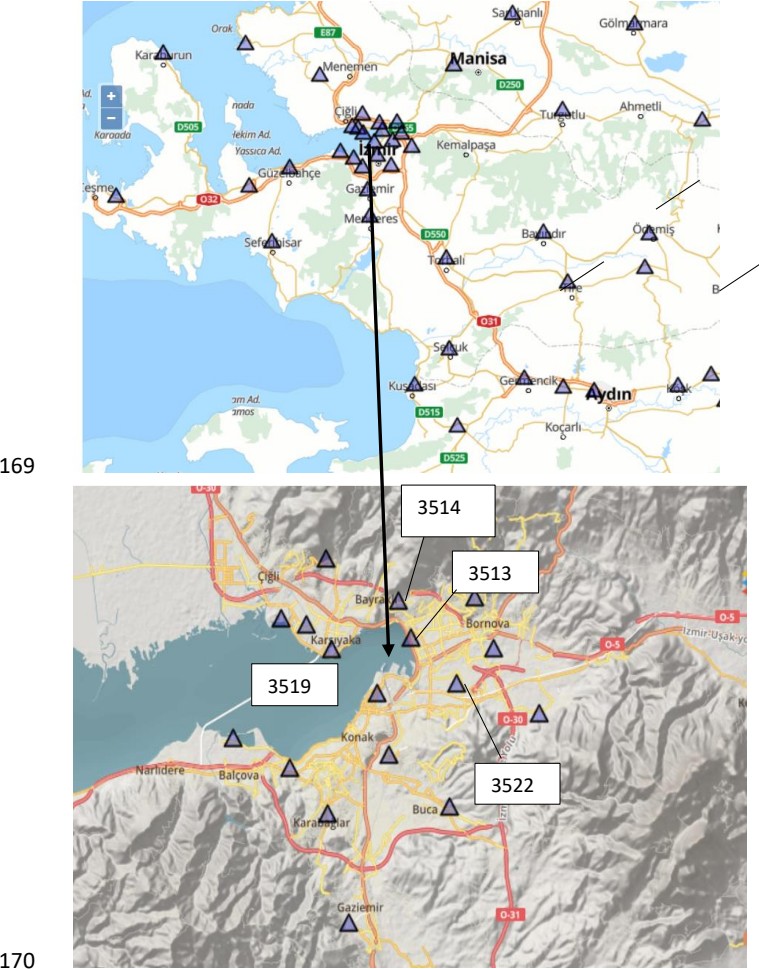


Figure 4. Overview of Stations in Izmir
The procedure was to use and select an appropriate outcrop rock site and use its corresponding data to further
determine the site response analysis of the selected soil sites. The selected stations were given in Table 4, with the
corresponding location and PGA data's.  The outcrop station was selected as station 3514, which is very close to
the basin area. Geotechnical and geophysical properties of the selected stations are given in Figure 5.




高





Table.4 Selected Soil/Rock Sites

| **Site Name** | **Region** | **Vs30 (m/sec)** | **Coordinates** | **PGA (g)** |
|:---:|:---:|:---:|:---:|:---:|
| 3514 | Bayrakli | 836.00 | 38.4762 27.1581 | 0.057 (E-W) |
| 3513 | Bayrakli | 195.00 | 38.4584 27.1671 | 0.108 (N-S) |
| 3519 | Karsiyaka | 131.00 | 38.4525 27.1112 | 0.153 (N-S) |
| 3522 | Bornova | 249.00 | 38.4357 27.1987 | 0.075(E-W) |

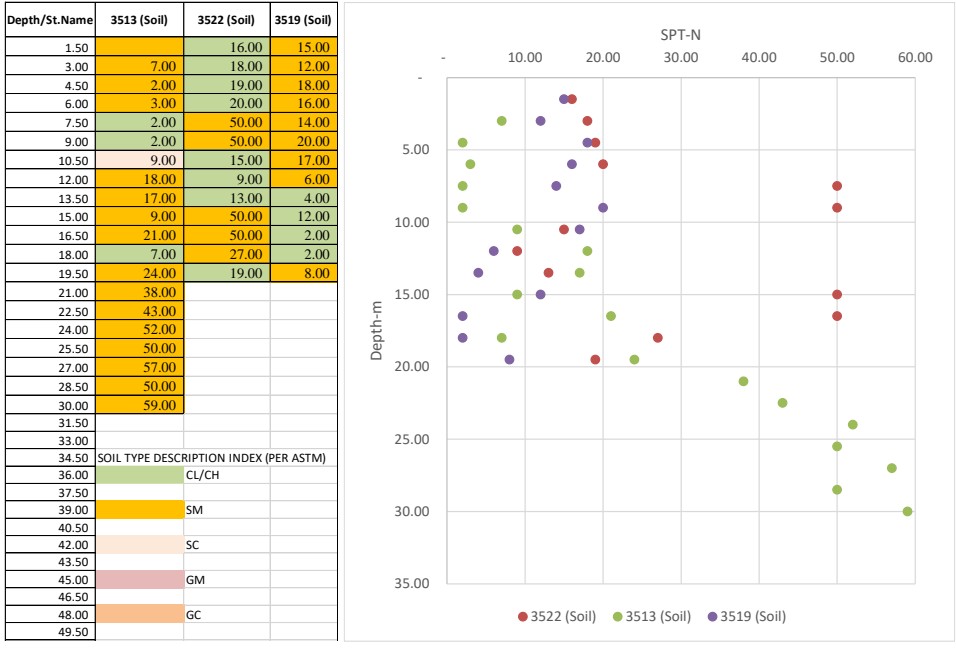




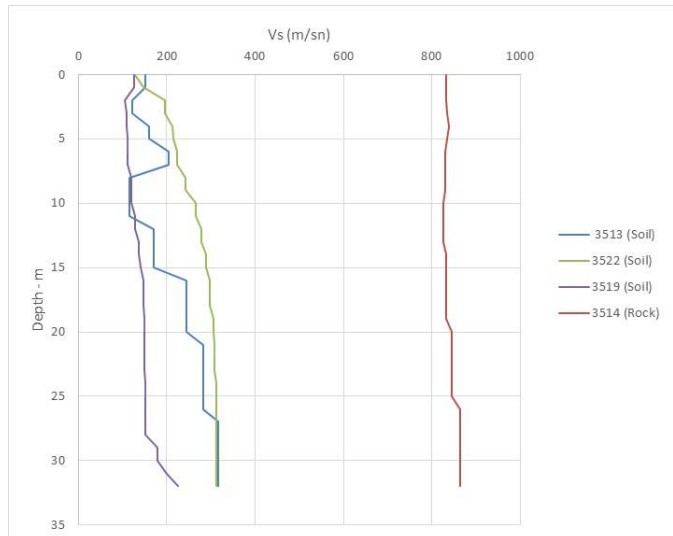


Figure 5. Geotechnical / Geophysical properties of the selected rock / soil stations
Detailed soil profiles and parameters were gathered from available deeper site profiles and deep geophysical
measurements that were used from the wide range of database. (Cetin et al, 2022) The modulus reduction and
damping curves were used from the literature by adopting soil parameters and the general trend of the curves were
given in Figure 6.
The results of the analysis were given for 3 different locations in the city center as previously stated. The motivation
for selecting 3 different regions was to take into account of different soil/geophysical properties of soils which
have alluvial soil deposits. The second motivation come from the fact that, to be able to generate a general response
spectrum, a more representative solution should be taken into account which represents the different soil conditions
and regions of the city (Figure 7-8-9).



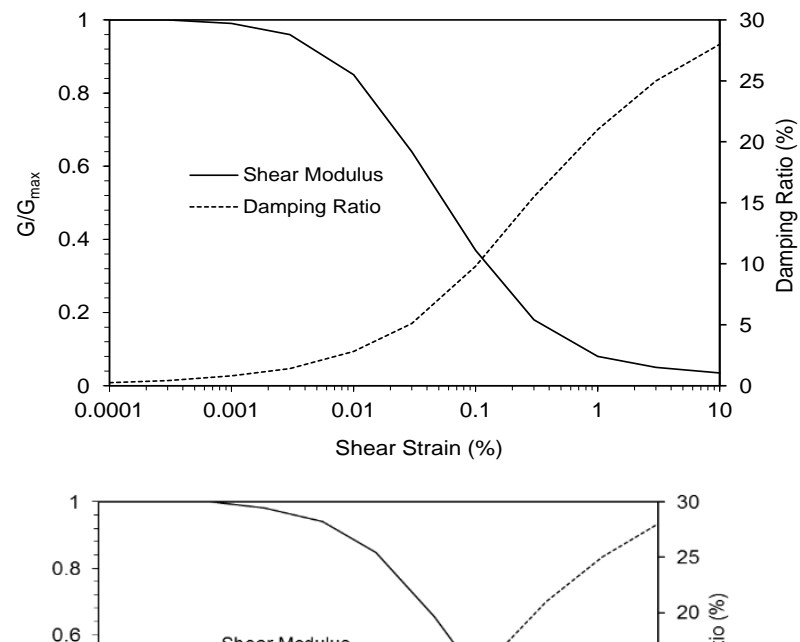



Figure 6. Modulus Reduction/Damping Curves for Sandy-Cohesionless Soils (Seed and Idriss, 1970) and
Cohesive Type Soils (Vucetic-Dobry, 1991)
In the results, comparison of the response spectrum graphs of SRA with the recorded site motion and corresponding
amplification function S_amp= SR (site) / SR (outcrop) were given for the 3 selected stations.  It can be seen that
the 3513 station amplifications increased up to 4 - 4.5 times in 1.50 s periods. Similar to 3513 station, at 3522
Bornova station, amplifications were observed for the same period region with 3.0 – 3.5 times increase.  These
two regions were close to each other and the geotechnical site conditions were similar to each other compared to
the Karsiyaka station. When the outcomes of Karsıyaka station (3519) was examined, similar amplification data
was obtained but in higher period regions, 2.50 seconds and later.



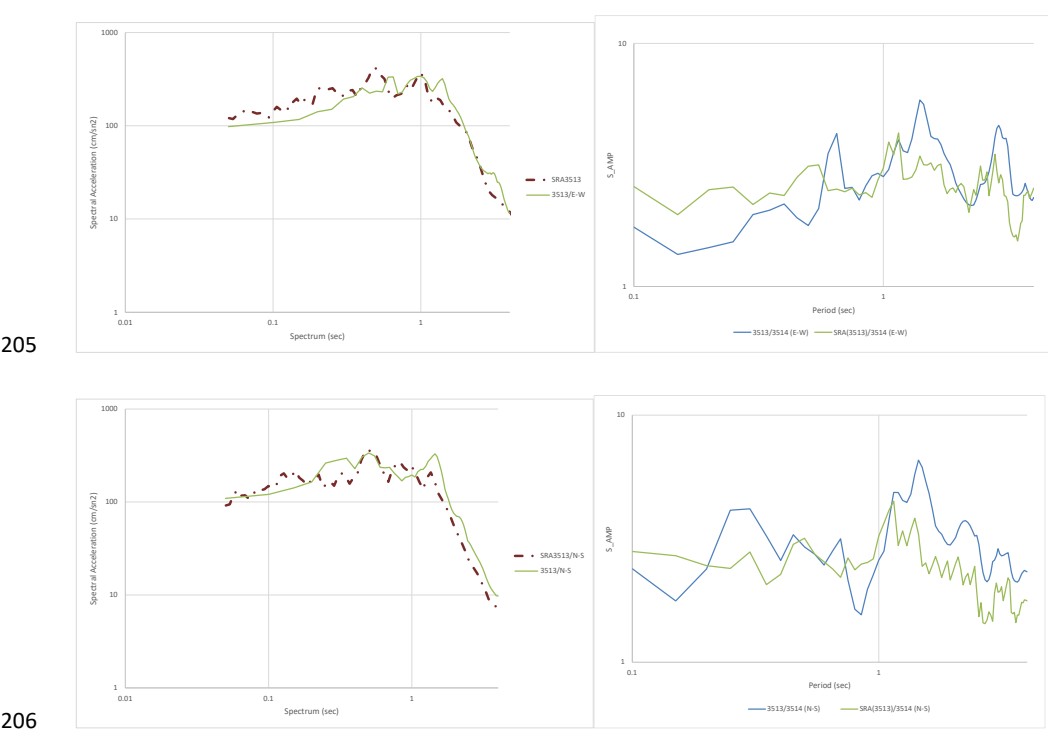



Figure 7. The comparison of recorded and estimated (SRA) elastic response and amplification spectra for 2020
Samos event@station 3513

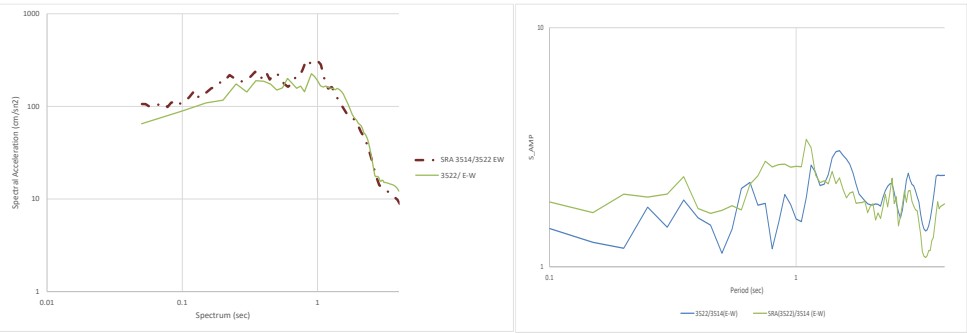




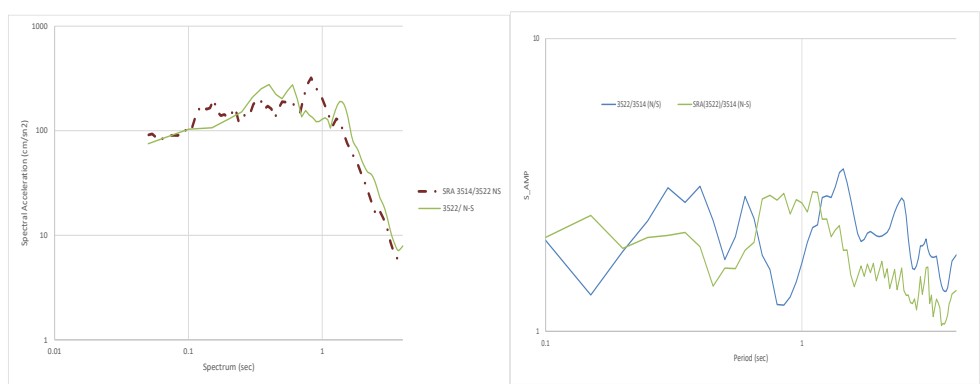


Figure 8. The comparison of recorded and estimated (SRA) elastic response and amplification spectra for 2020

Samos event@station 3522

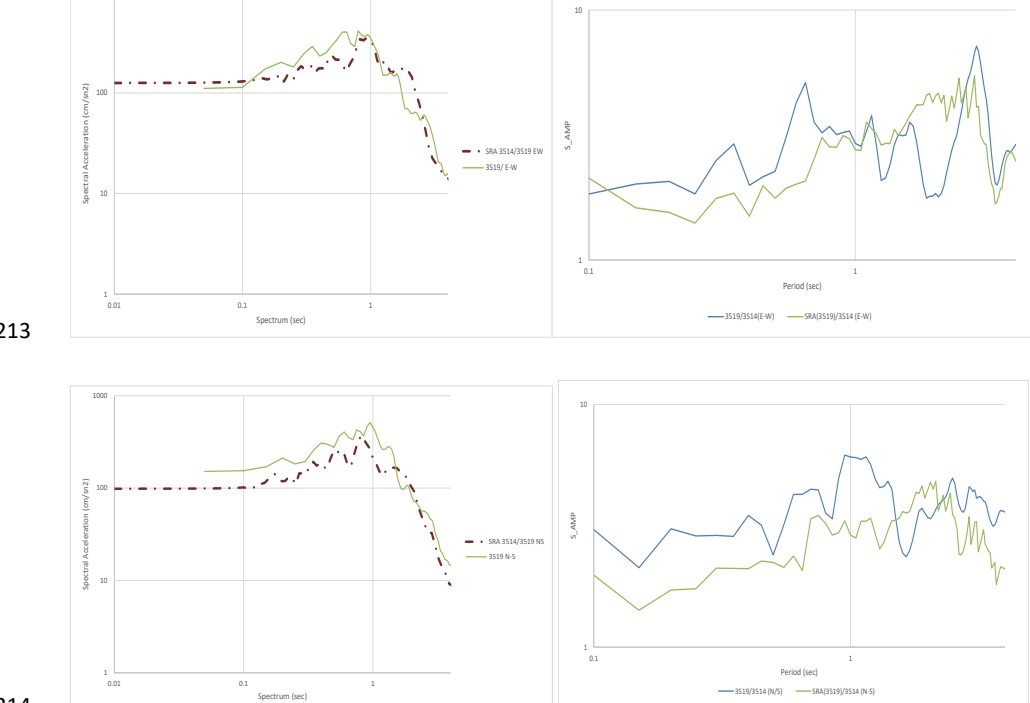



Figure 9. The comparison of recorded and estimated (SRA) elastic response and amplification spectra for 2020
Samos event@station 3519



The results indicate a conformity in the general trend of spectrum and have been found to be consistent with the
actual data especially in the period range of 0.50-1.50 sec range which coincides with the general building stock
(6-10 story heights) of the city. Overall, it can be concluded that, in most instances, the average spectra of the
recorded motions fall within the range of those associated with the calculated motions. The match between the
75th percentile of the recorded motions and the computed motions varies from moderate to very good. Similar
results can also been seen in Cetin et.al. (2024).
By evaluating the data and analysis results obtained so far, a reasonably usable SRA model and GMPE
relationships that can correspond to the seismicity of the general region have been revealed. The next stage will
be selecting the target spectrum for possible future earthquakes and then determining the spectral outcomes for
selected regions by performing SRA analyzes.
**4.  Selecting target response spectrum and evaluating the results of future events**
Using the most appropriate GMPE identified through the error analysis as stated before, ground motion parameters
such as Peak Ground Acceleration (PGA), Spectral Acceleration (SA), and others are predicted for a future
predicted deterministic scenario conditions. The target spectrum was generated for the deterministic scenario of
Radius Project (Radius, 1997) which was a detailed study for the seismicity of the region. The Project concluded
with a deterministic scenario which include an Mw 6.5 event in Izmir fault with an anticipated distance of 4 km.
An important consideration in site-specific seismic hazard analyses is the near-fault effect and the maximum
directional effect. Somerville et al. (1997) adjusted empirical ground motion attenuation models to account for the
influence of rupture directivity on both amplitude and duration. Rupture directivity happens when seismic energy
is concentrated along the path of fault rupture, leading to a substantial increase in ground motions in that direction.
This phenomenon is particularly significant for sites near faults, where rupture directivity can cause ground
motions to be considerably stronger in one direction, especially at longer periods, compared to others. This
behaviour can be observed in 2000 Samos event after investigating the N-S and E-W spectrums. The directivity
of the fault enhance the motions in N-S directions, which can also be associated with the damage behaviour of
buildings in Mavisehir- Karsiyaka region, specifically at station 3519. Therefore, this effect has been considered
in future earthquake simulations as well and the selected GMPE were revised accordingly. Target spectrum was
selected taking into account that the TEC and GMPE spectrums will not be underscored at ant period point.
Therefore, a new spectrum is generated which takes into account of the historical seismicity of the region as well
as the current regulations (Figure 10).





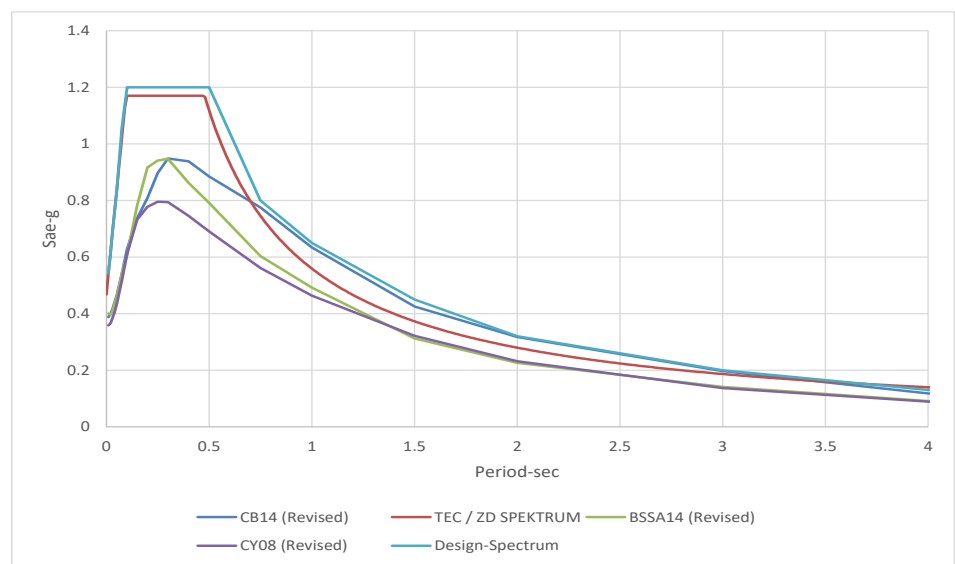


Figure 10. Design Spectrum for the site with comparisons
There are a total of 11 records were selected (Table 5) and scaled to the given target spectrum (Figure 11). The
scaling of the records are generated through Seismosoft software. The results of the selected and scaled ground
motions are given together in Figure 11.
Table 5. Selected Ground Motion Records

| No | Record Sequence Number | Scale Factor | Earthquake Name | Year | Station Name | Magnitude | Mechanism | Rjb (km) | Rrup (km) | Vs30 (m/sec) |
|---|---|---|---|---|---|---|---|---|---|---|
| 1 | 4881 | 2.32 | "Chuetsu-oki_Japan" | 2007 | "Nagaoka Kouiti Town" | 6.8 | Reverse | 11.61 | 20.77 | 294.38 |
| 2 | 549 | 1.91 | "Chalfant Valley-02" | 1986 | "Bishop - LADWP South St" | 6.19 | strike slip | 14.38 | 17.17 | 303.47 |
| 3 | 6893 | 1,07 | "Darfield_New Zealand" | 2010 | "DFHS" | 7 | strike slip | 11.86 | 11.86 | 344.02 |
| 4 | 8133 | 4.31 | "Christchurch_New Zealand" | 2011 | "SLRC" | 6.2 | Reverse Oblique | 31.81 | 31.81 | 249.28 |
| 5 | 6971 | 2.12 | "Darfield_New Zealand" | 2010 | "SPFS" | 7.0 | strike slip | 29.86 | 29.86 | 389.54 |
| 6 | 882 | 2,57 | "Landers" | 1992 | "Desert Hot Springs" | 7.28 | strike slip | 26.84 | 26.84 | 344.67 |
| 7 | 4866 | 1.26 | "Chuetsu-oki_Japan" | 2007 | "Kawanishi Izumozaki" | 6.8 | Reverse | 0.0 | 11.75 | 338.32 |
| 8 | 4894 | 0.38 | "Chuetsu-oki_Japan" | 2007 | "Kashiwazaki NPP_Unit 1: ground surface" | 6.8 | Reverse | 0.0 | 10.97 | 329.0 |
| 9 | 787 | 1.43 | "Loma Prieta" | 1989 | "Palo Alto - SLAC Lab" | 6.93 | Reverse Oblique | 30.62 | 30.86 | 425.3 |
| 10 | 1100 | 2.03 | "Kobe_Japan" | 1995 | "Abeno" | 6.9 | strike slip | 24.85 | 24.85 | 256.0 |
| 11 | 3979 | 2.82 | "San Simeon_CA" | 2003 | "Cambria - Hwy 1 Caltrans Bridge" | 6.52 | Reverse | 6.97 | 7.25 | 362.42 |






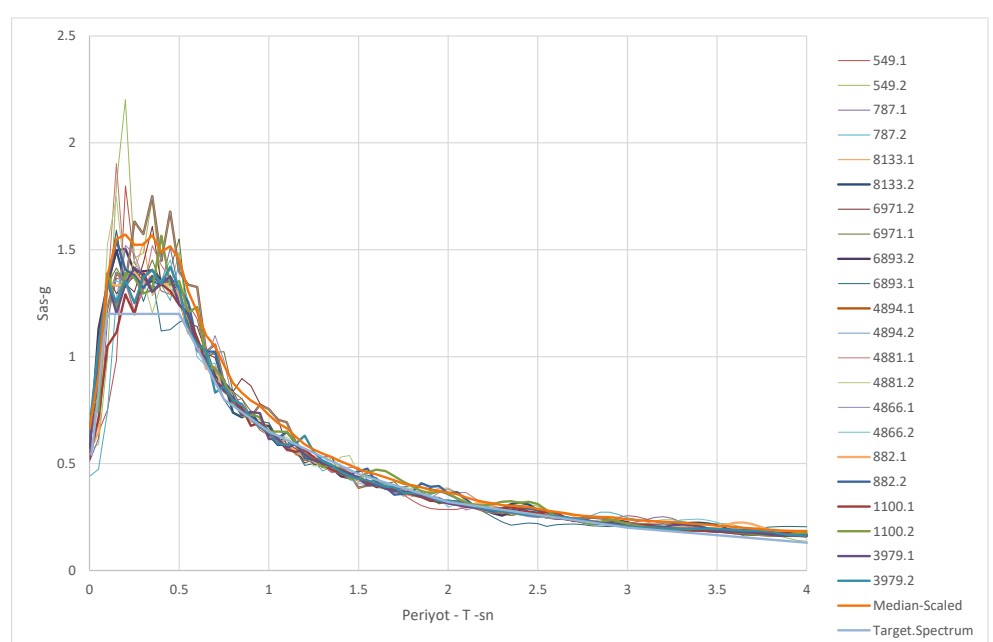

Figure 11. Selected and Scaled Ground Motion with respect to target spectrum

Using the deepsoil model calibrated in previous sections, site-specific earthquake analyzes with selected records were carried out for each region/station and results were given in Figure 12.

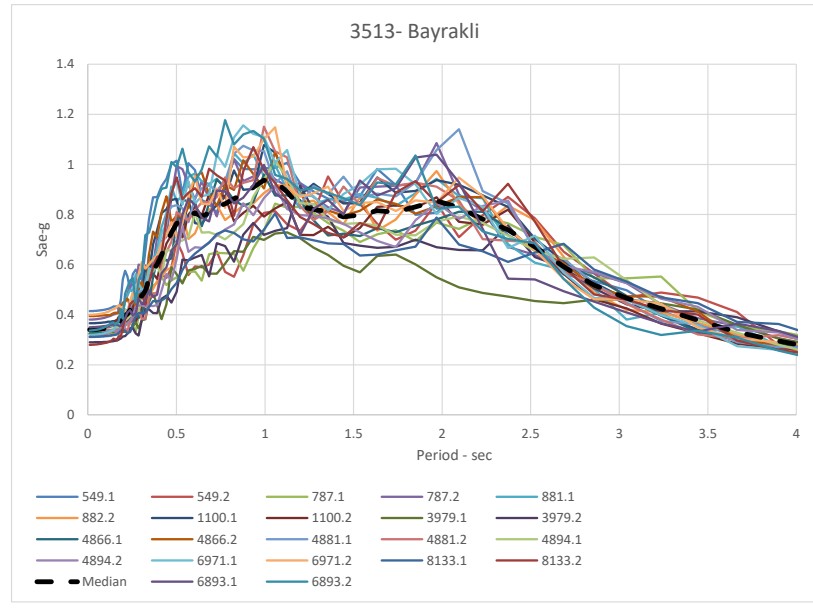



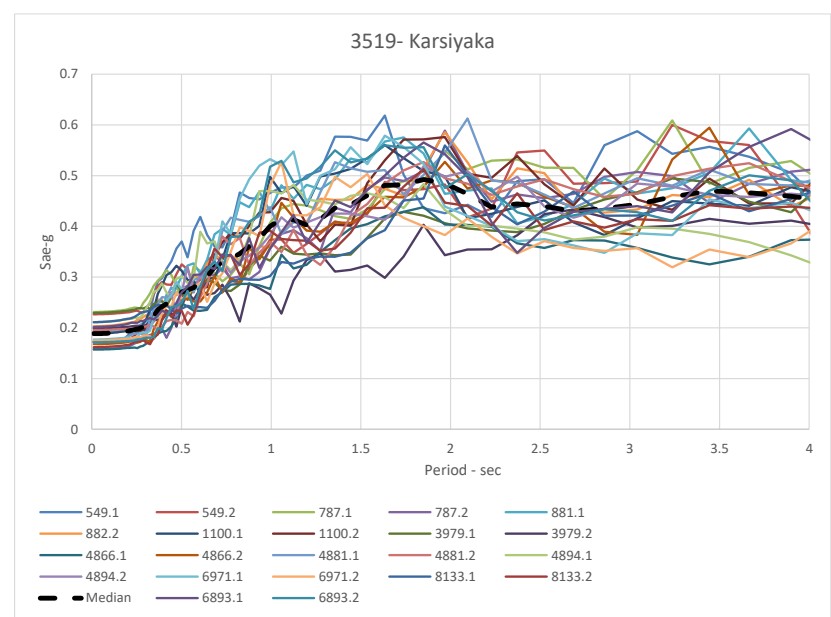


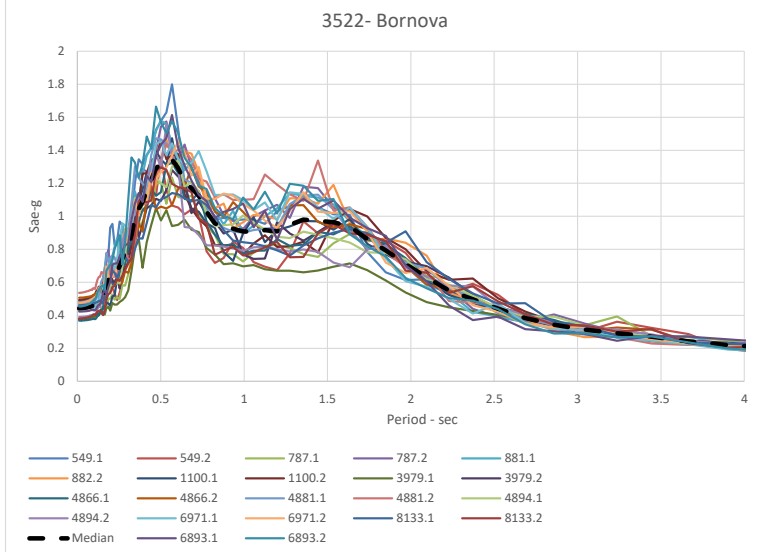


Figure 12. Outcome of the future anticipated scenario earthquake in the city with 3 different regions (Bayrakli,
Karsiyaka and Bornova)
The comparison of each region with the current TEC and target spectrum were given in Figure 13.



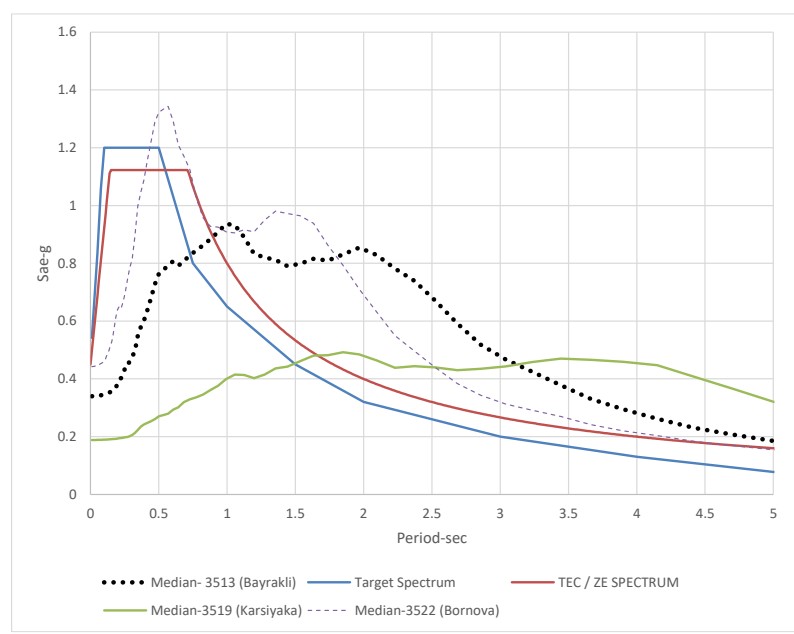


Figure 13. The comparison of stations with the current TEC and target spectrum
According to TEC, local site effects are taken into account by some modification
factors. These modification factors are called $F_1$ and $F_S$ values and defined by the following relationship:
Sds = $S_S F_S$   (Sds = Design Spectral Acceleration Value for short period region)
Sd1 = $S_1 F_1$ (Sd1 = Design Spectral Acceleration Value for 1sec period region)
Where $S_{S \text{ and }} S_1$ are the spectral values without taking into account the local site effects.
The result of the analysis showed that the local site modification factors should be corrected by at least two times
as summarized in Table 6.
Table 6. Local site modification factors (F1) according to TEC and SAR of scenario earthquake

| Station | Vs30 | Soil Class (Acc TEC) | F1/ TEC | F1/ SAR, t=1 sec |
|---------|------|----------------------|---------|-------------------|
| 3513 | 195 | ZD | 2.054 | 4.21 |
| 3522 | 249 | ZD | 2.054 | 4.76 |
| 3519 | 131 | ZE | 2.935 | 4.40 |








### 5. Summary and Conclusions

In this study, based on the past seismicity of the city of Izmir, potential future seismicity of the city of city has been considered and various analyses have been conducted. A summary of these studies is provided below.

- Firstly, using a dataset of past recorded earthquake events, the level of agreement with current GMPE equations was investigated. Based on the evaluations, two GMPEs were selected for use in determining target spectrum parameters for site-specific seismicity analysis.
- To perform site-specific seismic analyses, well-recorded event of İzmir-Samos earthquake data were utilized. A 1D analysis model, using the available geotechnical data was applied for 3 different stations. These stations were selected for the aim to
  - represent different alluvial soil conditions of the city
  - take in to account of the 3 most populated, therefore representative regions of the city
  - be able to arrive a more general conclusion about the possible future earthquake simulations
- Future potential earthquake scenarios have been selected. For this purpose, a target spectrum was developed for an Mw=6.5 earthquake on the İzmir fault, as part of the RADIUS 2005 project. The resulting target spectrum was modified to account for near-field and directivity effects and subsequently used in the analyses.
  - The TEC was also utilized in the selection of the target spectrum. Ultimately, the chosen target spectrum was developed to satisfy both deterministic and probabilistic approaches given by the code recommendations.
- Eleven earthquake records were selected and scaled to match the target spectrum. Subsequently, using the same validated models, possible scenario earthquake outcomes were analyzed.

The results obtained from the analyses are provided below:

- The 2020 Samos earthquake has been a significant event for site-specific seismicity studies due to the abundance of recording stations and the rich data content available. In the analyses conducted, amplifications were observed in the high-period region.
- GMPEs were evaluated and compared using the past seismic activity of the city. That further allow to generate a target spectrum for the city.
- While generating a target spectrum, particularly when considering near-field effects and directivity effects, the obtained spectra possess a broader energy content than those presented in the regulations. This condition should be taken into account in seismic design codes.
- The acceleration spectra obtained at the surface are amplified by at least a factor of 2 for periods of 1 second and longer. More specifically, the result of the analysis showed that the local site modification factors defined by TEC should be corrected by at least 2.50 times. This condition should be taken into account in all of the alluvial regions of the city especially when designing more than 8-10 stories of buildings.
- Different regions selected in this study provides a framework for a future study which will emphasize on the basin effect discussed in other papers.



**Funding** The authors declare that no funds, grants, or other support were received during the preparation of his
manuscript.
**Data availability** all data and models analyzed during the current study are available from the corresponding
author on reasonable request.
**Declarations**
**Competing interests** the authors have no relevant financial or non-financial interests to disclosure.

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
