# Peer review of "Critical Evaluation of Strong Ground Motions in Izmir and Implications for Future Earthquake Simulation"

_EGUsphere, 2024_

## Author Comment (AC1)

27.07.2025

Response to Reviewer Comments

Manuscript ID: EGUsphere-2024-3488

Title: Critical Evaluation of Strong Ground Motions in Izmir and Implications for Future Earthquake Simulation Results

Dear Editor and Reviewers,

We sincerely thank the anonymous reviewer(s) and the handling editor for their constructive and insightful feedback on our manuscript. In response, we have carefully revised the manuscript and addressed all comments in detail. The updated version incorporates several notable improvements:

- A clearer articulation of the study's novelty and regional significance has been added to the Introduction.
- The structure of the manuscript has been refined, including the separation of seismotectonic background into a dedicated section for improved flow and clarity.
- All figures and tables have been revised for resolution, readability, and consistency.
- Quantitative validation metrics have been introduced to support the site response analysis.
- Discussions on GMPE performance, site amplification, and directivity effects have been expanded and clarified in line with reviewer suggestions.

Reviewer comments are shown in blue, while our responses are shown in black. In the revised manuscript, all changes made in response to reviewer comments are highlighted in yellow for ease of reference.

We believe that these revisions have significantly strengthened the manuscript both in content and presentation. The revised version now presents a clearer, more coherent, and technically sound contribution that aligns with the scope of *Natural Hazards and Earth System Sciences (NHESS)*.

Please find below our detailed, point-by-point responses to the reviewer's comments.

Respectfully,

**Şahin Çağlar Tuna**

**General Evaluation**

This manuscript addresses a highly relevant and practical problem: evaluating strong ground motions in Izmir and implications for future earthquake simulations using GMPEs, site response analyses (SRA), and comparison with Turkish Earthquake Code (TEC) design spectra. The study includes an extensive dataset, thoughtful site-specific calibration, and a compelling case for re-evaluating seismic design spectra for alluvial zones in Izmir. The work is timely and important, particularly in the context of recent seismic events like the 2020 Samos Earthquake.

Although the manuscript requires several clarifications and presentation improvements, these revisions do not affect the core methodology or conclusions. Therefore, I recommend acceptance after minor revision.

**General Comments:**

These are not major scientific flaws, but addressing them will improve the clarity, quality, and impact of the paper:

1. Clarify Novelty and Contribution

A short paragraph clearly stating how this study builds upon or extends those provious works (e.g., broader dataset, new scenario, comparison with TEC, etc.) would be helpful.

**Author's Response:** Thank you for the comment. In response, we have added a concise paragraph to the end of the *Introduction* section (Lines 79–83 in the revised manuscript). This addition emphasizes how the current study extends prior work by integrating a longer-term strong motion dataset (1996–2024), performing error-based GMPE performance assessments, and applying a calibrated DeepSoil model to simulate site-specific future earthquake scenarios. Moreover, the study uniquely compares amplification outcomes with the current Turkish Earthquake Code (TEC, 2018), highlighting the potential underestimation of seismic demands in alluvial zones. We trust that this clarification effectively addresses the reviewer's concern and improves the clarity of our manuscript's scientific contribution.

2. Improve Figure Quality: Some plots (e.g., Figures 7–13) suffer from low resolution, small font sizes, and unclear legends. It is recommended to: Increase resolution to 300 dpi, Use consistent color schemes and line types, Clearly label axes and add proper legends, Language Polishing. Minor grammatical errors and awkward phrasing (e.g., "data riched event") should be corrected throughout the manuscript. A careful proofreading or light professional editing is encouraged.

**Author's Response:** Thank you for the comment. In response:

- All figures mentioned (Figures 7–13) have been revised and updated and improved font readability, and consistent color schemes and line styles. Axis labels and legends have also been refined for clarity and uniformity across the manuscript.
- The manuscript has undergone a thorough proofreading process. Specific issues such as the phrase *"data riched event"* and other minor grammatical errors have been corrected. We believe these revisions improve both the visual quality and overall readability of the paper.

3. Enhance Discussion of Uncertainty: While the GMPE selection and SRA analyses are rigorous, a brief paragraph discussing epistemic uncertainties (e.g., in soil properties, GMPE selection) would strengthen the credibility of the conclusions.

**Author's Response:** Thank you for the comment. In the revised version, we have included a new paragraph at Section 2 between lines 151-156 to explicitly address these sources of uncertainty. The paragraph highlights the limitations associated with regional GMPE applicability, simplified site characterization through Vs30, and near-fault data scarcity. We believe this addition improves the transparency and scientific rigor of the analysis.

4. Cite Recent Hazard Models or Studies for İzmir: Since the RADIUS project dates back to 1997, it would be useful to mention whether newer seismic source models or hazard assessments (e.g., AFAD, SHARE) have been compared or considered.

**Author's Response:** Thank you for the comment. In the revised manuscript, we have incorporated a justification for the continued relevance of the Mw = 6.5 scenario originally proposed in the RADIUS (2005) study. Specifically, we added a paragraph in the summary section referencing recent seismic source characterizations from the EFSM20 (Basili et al., 2024) and ESHM20 (Danciu et al., 2024) models. According to EFSM20, the İzmir Fault is classified as an active crustal fault with a maximum moment magnitude potential of Mw 6.5–6.7. In parallel, spatial recurrence maps from ESHM20 (see Fig. 3c–d) indicate annual rates of $\log_{10}[\text{Rate}] \approx -6.5$ to $-7.0$ for Mw > 6.5 events across the İzmir basin. These data confirm the plausibility and consistency of the selected deterministic scenario with the most up-to-date regional hazard models. The manuscript has been updated accordingly, and full references have been added.

5. I recommend moving Figure 1 and the related tectonic context from the Introduction into a separate section (e.g., "Seismotectonic Background"). Given its importance for ground motion characteristics and scenario development, the seismogenic framework deserves clearer, dedicated presentation rather than being embedded in the general introduction.

**Author's Response:** Thank you for the comment. In the revised version of the manuscript, we have created a new Section 1.1 – "Regional Seismicity and Geological Settings of Izmir Bay" to explicitly present the seismotectonic context. This section now includes a detailed discussion of the regional tectonic regime, active fault systems, and geological conditions of Izmir, supported by updated Figure 1 and Figure 2. The section clarifies the role of extensional tectonics, the influence of nearby fault systems (e.g., İzmir and Seferihisar Faults), and the amplification potential associated with alluvial basins and soft sediments. These elements provide the necessary foundation for the scenario-based seismic hazard analyses conducted later in the study.

**Detailed comments by section**

**Section 1.1 (Scope and Aim):**

6. I recommend relocating Figure 1 and the associated tectonic context into a new subsection (e.g., "Seismotectonic Background"), as this content forms the geological basis for the entire study.

**Author's Response:** Thank you for the comment. We thank the reviewer for this valuable suggestion. In response, we have reorganized the manuscript structure by introducing a new subsection titled "1.1 Regional Seismicity and Geological Settings of Izmir Bay" under the Introduction. The tectonic context, geological description, and Figure 1 have been relocated to this section. This restructuring improves the logical flow of the manuscript and more clearly highlights the geodynamic framework that underpins the study. We agree that this change better reflects the foundational role of regional seismotectonics in the subsequent analyses.

7. The introductory paragraphs are overly general. I suggest revising them to emphasize specific regional seismic characteristics (e.g., fault segmentation, recurrence, basin amplification) supported by references.

**Author Response:** Thank you for the comment. In response, the Introduction section has been substantially revised to focus more specifically on the unique regional seismic characteristics of İzmir. The following improvements have been made:

- Basin amplification effects have been explicitly discussed, especially referencing the 2020 Samos earthquake and observed exceedances of the Turkish Earthquake Code (TEC, 2018) by factors of 4–6 in the 0.7–1.0 s period range (Cetin et al., 2022; Gülerce et al., 2022).
- The underlying Quaternary alluvial basin geometry, soft soil profiles, and impedance contrasts are now identified as primary contributors to spectral amplification, in line with recent findings (Cetin et al., 2023).

- The need for regionally calibrated GMPEs, and limitations of current site classification-based spectra in capturing spatial seismic demand variability across İzmir Bay, are now clearly articulated.
- Finally, the introduction now outlines a more detailed and structured study purpose and methodology, emphasizing deterministic modeling, residual-based GMPE analysis, and site response validation in basin environments.

These changes are aimed at addressing the reviewer's concern by emphasizing localized seismotectonic and geotechnical conditions that drive the study's motivation. The revised introduction now provides a more technically grounded and regionally specific context for the research.

We believe these revisions have substantially strengthened the scientific framing of the study and aligned it more closely with the journal's scope and the reviewer's expectations.

8. References such as Emre et al. (2018) and McKenzie (1978) should be more clearly contextualized—why they are relevant and how they support the regional hazard description.

**Author Response:** Thank you for the comment. In the revised manuscript, the references to Emre et al. (2018) and McKenzie (1978) have been expanded and clarified to emphasize their relevance.

- **Emre et al. (2018)** is now explicitly cited as the primary source for the most recent and detailed segmentation of active faults in Turkey, including the Izmir and Seferihisar fault systems. Their GIS-based fault database defines key fault parameters (e.g., length, slip rate, activity class), which underpin the regional seismic hazard framework.
- **McKenzie (1978)** has been contextualized as the foundational work identifying the extensional tectonic regime in the Aegean and its relationship with subduction along the Hellenic arc. This study offers geodynamic context for the crustal stretching and normal faulting observed in Western Anatolia, which directly affects Izmir's seismicity.

These clarifications are now incorporated into Section 1.1.

9. Minor language edits are needed (e.g., "data-riched" → "data-rich"; restructure awkward phrases for clarity).

**Author Response:** Thank you for the comment. A careful proofreading has been performed throughout the manuscript, and all identified errors and awkward phrases have been revised for clarity and accuracy. For example, the phrase *"data-riched"* has been corrected to *"data-rich"*, and other expressions have been restructured to improve fluency, readability, and professional tone. These changes appear throughout the revised version of the manuscript.

Section 1.2 (Geological and Geotechnical Settings of İzmir Bay):

10. This section provides useful context; however, the distinction between geological and geotechnical properties is vague. Consider separating lithological description (e.g., rock types, basin structure) from engineering properties (e.g., Vs30, soil classes, amplification potential).

**Author Response:** Thank you for the comment. In response, we revised Section 1.1 to focus exclusively on the geological and seismotectonic framework, as reflected in the updated title (*"Regional Seismicity and Geological Settings of Izmir Bay"*). Portions of the seismological context—such as fault segmentation, recurrence intervals, and tectonic regime—previously located in the Introduction have been relocated to this section for better thematic coherence.

Meanwhile, the geotechnical site conditions, including soil classifications, Vs30 values, and amplification characteristics, are presented in detail in Section 3, where they are directly integrated into the analytical modeling and site response analyses. This separation improves clarity and ensures that technical content is introduced at the appropriate depth within the structure of the manuscript.

11. Language could be improved for precision: "Loosely consolidated sediments can exacerbate ground shaking" → "Loose alluvial soils with low stiffness can significantly amplify seismic waves, particularly in the 0.5–2 s period range."

**Author Response:** Thank you for the comment. The phrase has been revised as recommended to more accurately reflect the geotechnical behavior of the soil profile in lines 120-122.

Section 2: GMPE Dataset and Model Comparison

12. The inclusion of 33 real earthquake events with source and PGA data is commendable. However, please clarify how site conditions were accounted for (e.g., Vs30, site class) in the GMPE comparisons. Were all sites considered as rock, soil, or site-adjusted?

**Author Response:** Thank you for the comment. In this study, all 33 ground motion records were obtained from two strong motion stations located in Bornova, Izmir: Station 3502 (Vs30 = 270 m/s) and Station 3522 (Vs30 = 249 m/s). Both stations are classified as NEHRP Site Class D. The use of two stations with consistent site classification minimizes variability due to soil effects and enhances the comparability and reliability of the GMPE performance evaluation. This also improves the applicability of the results to typical soft soil conditions found across urban areas of Izmir.

To account for site amplification in the GMPE comparisons, the actual Vs30 values for each station were entered into the models using the site adjustment formulations specific to each GMPE. This ensures a consistent and model-compliant estimation of PGA values. Additionally, the selected dataset includes records spanning a broad range of earthquake magnitudes and source-to-site distances, thereby providing a more comprehensive basis for evaluating the robustness and generalizability of the GMPEs.

This clarification has been added to the manuscript in Section 2 (Lines 151–161).

13. Table 1 is rich, but consider: Reordering events chronologically or grouping by distance/magnitude for better pattern recognition. Adding a column for site class or Vs30 (if available).

**Author Response:** Thank you for the comment. In the revised manuscript, Table 1 has been reordered chronologically to enhance readability. Site class and Vs30 information for the two recording stations used in this study is already provided in the main text (lines 146-150). Therefore, this information was not repeated in the table.

14. Table 2 (GMPEs) is well-structured, but the rationale for including only NGA-West1 and NGA-West2 models should be briefly explained. Are there any Turkish/regional GMPEs that were excluded, and if so, why?

**Author Response:** Thank you for the comment. The selection of GMPEs from the NGA-West1 and NGA-West2 projects was based on their broad magnitude–distance coverage, well-defined site adjustment terms (e.g., Vs30-based amplification), and wide international acceptance. These models have been extensively validated and are supported by open-access coefficient sets, which allow for consistent implementation in site-specific analyses.

While several regional GMPEs have been developed for Turkey and the broader Euro-Mediterranean region (e.g., Akkar & Bommer, 2010; Akkar et al., 2014; Kalkan & Gülkan, 2004), they were not included in this study due to various limitations—such as restricted magnitude ranges, limited near-fault recordings, or lack of detailed site adjustment coefficients. Nonetheless, we acknowledge their relevance and consider their inclusion in future regional comparisons as a valuable direction. A brief explanation has been added to the manuscript in Section 2 (Lines 130–141).

15. Equation formatting (RMSE) is clear, but please define variables (e.g., N, i) inline or as subscript to improve readability.

**Author Response:** Thank you for the comment. We thank the reviewer for this helpful comment. In response, the RMSE equation has been revised for improved clarity, and all variables ($N$, $i$, $PGA_{observed,I}$, $PGA_{predicted,I}$ are now clearly defined using consistent inline and subscript notation. These corrections appear in Section 2 (Lines 174–177) of the revised manuscript.

16. Figure 3 (scatter plots): R² values are useful, but consider adding residual plots or at least comment on residual trends (e.g., underprediction for short distances or large PGA?). Improve axis labeling (units missing or too small).

**Author Response:** Thank you for the comment. In response:

- We have expanded the original analysis by including residual plots for the two top-performing GMPEs (CB14 and BSSA14), which are now presented as Figure 4a and 4b.
- These plots demonstrate systematic prediction biases:

CB14 exhibits underprediction trends at short epicentral distances and for higher observed PGA values, likely due to unmodeled near-fault or local amplification effects.

BSSA14, on the other hand, shows a broader pattern of overprediction, particularly for low-magnitude or far-field events.

- We also revised Figure 3, which now presents observed vs. predicted PGA scatter plots for all eight GMPEs. The figures include:
  - Clearly labeled axes with appropriate units,
  - A solid red 1:1 reference line,
  - A blue dashed regression line, and
  - Inline R² annotations in the upper-left of each panel, with legends repositioned to avoid overlap.

These updates have been integrated into Section 2 and discussed accordingly. The revised figures allow for clearer interpretation of both overall model performance and residual trends, as recommended.

17. The error metrics in Table 3 clearly identify CB14 and BSSA14 as the most appropriate models. Still, a brief sentence on whether residual bias was directionally consistent across magnitude or distance ranges would strengthen the justification.

**Author Response:** Thank you for the comment. As suggested, we have added a clarifying statement regarding the directional trends in residuals:

Residual plots for CB14 and BSSA14 indicate that model-specific biases are not random but directionally consistent. CB14 exhibits systematic underprediction at short epicentral distances and for higher magnitude events, while BSSA14 shows a general overprediction trend across low-magnitude and far-field records.

This observation has been incorporated into Section 2 (Lines 193-204) to support the statistical justification for selecting these two models.

Section 3: Site Response Validation

18. The use of the 2020 Samos earthquake for model calibration is appropriate. However, the phrase "data-riched event" should be corrected to "data-rich event.

**Author Response:** Thank you for the comment. In response, the phrase "data-riched event" has been removed. Instead of directly using this expression, we have restructured the related sentence to more appropriately convey the richness and reliability of the 2020 Samos earthquake dataset within the revised context of Section 3. This ensures clearer and more accurate academic language while preserving the intended meaning. We appreciate the reviewer's attention to linguistic precision.

19. The selection of stations (3513, 3519, 3522) is logical and well-supported. Still, please clarify the criteria for choosing station 3514 as the outcrop reference. Is it based on rock classification by Vs30? A sentence confirming this would help.

**Author Response:** Thank you for the comment. To address your suggestion, we have clarified the basis for selecting Station 3514 as the outcrop reference. Specifically, this station was chosen due to its high average shear-wave velocity ($Vs_{30}$ = 836 m/s)**,** which clearly classifies it as rock site according to NEHRP and Eurocode site classification systems. A confirming sentence has been added to Section 3 (Line 229-231) for clarity.

20. Tables 4 and Figure 5 provide essential input data, but a clear indication of soil class (e.g., ZC, ZD, ZE) and correlation with TEC site categories would enhance interpretation.

**Author Response:** Thank you for the comment. In the revised manuscript:

- Figure 5 has been updated (Figure 6 in revised manuscript) with improved visual resolution and clarity, presenting detailed SPT-N and Vs profiles.
- Table 4 has been revised to include an additional column showing the soil classifications (ZB, ZD, ZE) based on measured $Vs_{30}$ values in accordance with TEC (2018).
- Furthermore, lithological information describing the interbedded and transitional soil formations (CL/CH, SM, SC) has been incorporated into the main text (lines 240-246) for improved geological interpretation.

These enhancements now allow for a clearer understanding of local site conditions and their implications in seismic site response modeling.

21. The geotechnical profile descriptions and the use of modulus reduction/damping curves (Figure 6) are acceptable. However, it would be beneficial to state whether site-specific laboratory curves were available or if literature-based curves (Seed & Idriss, Vucetic & Dobry) were adopted uniformly.

**Author Response:**

**Author Response:** Thank you for the comment. In this study, well-established empirical models were deliberately adopted to represent the nonlinear behavior of the subsurface soils, given the absence of site-specific laboratory-based modulus reduction and damping ratio curves. Seed and Idriss (1970) relationships were utilized for cohesionless sandy soils, while Vucetic and Dobry (1991) curves were applied for cohesive soils, based on their plasticity characteristics. The selection was guided by detailed geotechnical profiling, including SPT-N values, Vs measurements, and soil classifications.

In accordance with the reviewer's suggestion, the methodology section has been revised to clearly explain this modeling choice. Additional references—such as Zalachoris and Rathje (2015)—have been incorporated to further support the validity of using literature-based nonlinear parameters in the absence of laboratory-specific data. These clarifications appear in Lines 253-257 of the revised manuscript.

22. Amplification results (SRA/Outcrop) are well presented (Figures 7–9), but legends and axes in the plots should be improved for clarity (e.g., distinguish observed vs. modeled more clearly).

**Author Response:** Thank you for the comment. Figures 7–9 have been revised to improve visual clarity. The legends have been updated to more clearly differentiate between observed and simulated results (e.g., SRA vs. recorded), using

distinct line styles and colors. Axis labels have also been enhanced for better readability, and annotation formats were standardized across all subplots. These updates help improve the interpretability of the results without altering the technical content.

23. The statement that amplifications match well between 0.5–1.5 s is important (lines 218–220) and should be connected to predominant building periods in İzmir (e.g., 6–10 story RC buildings) more explicitly.

**Author Response:** Thank you for the comment. Thank you for the helpful suggestion. The connection between the observed amplification in the 0.5–1.5 s period range and the predominant mid-rise reinforced concrete building stock in İzmir (6–10 stories) has been clarified in the revised manuscript. This revision emphasizes the engineering relevance of the site response results and their implications for seismic risk in the city. Please refer to Lines 293.297 in the revised version.

24. It would strengthen the validation to include a quantitative metric (e.g., goodness-of-fit score, misfit index, or confidence range) to support the visual matching of recorded and simulated response spectra.

**Author Response:** Thank you for the comment. We appreciate the reviewer's valuable suggestion regarding the need for quantitative support in validating the spectral matching between recorded and simulated responses.

As recommended, we have included four widely accepted quantitative metrics to assess the agreement between the computed and recorded spectral acceleration values:

– Root Mean Square Error (RMSE),– Coefficient of Determination ($R^2$),– Goodness-of-Fit Index (GOF), and – Nash–Sutcliffe Efficiency (NSE).

These metrics were computed for all six ground motion components analyzed in the study. The results are now presented in Table 5 of the revised manuscript and discussed in the section. Overall, the values confirm a good level of agreement between recorded and simulated spectra, particularly for components 3513e-w and 3519e-w, with $R^2 >$ 0.73 and GOF > 0.84.

We believe that the inclusion of these quantitative indices substantially improves the robustness and clarity of the validation process, and we thank the reviewer again for this constructive recommendation.

Section 4: Target Spectrum and Future Scenario Analysis

25. The inclusion of near-fault effects and rupture directivity (e.g., Somerville et al., 1997) is a strong contribution. However, this discussion (lines 234–246) could benefit from a short numerical explanation or quantification of how much this effect modifies the spectrum (e.g., amplification factor at 1–2 s).

**Author Response:** Thank you for the comment. In response to this valuable suggestion, we have revised the relevant paragraph in Section 4 to include a specific numerical example based on the 2020 Samos earthquake recordings into the revised manuscript in Lines 341–346.

This quantification provides a clear illustration of how rupture directivity significantly amplifies long-period ground motions, especially for near-fault sites aligned with the rupture path. These values are consistent with previous findings in the literature (e.g., Shahi & Baker, 2011; Bayless et al., 2024), and we have now cited these studies in support of our observation.

We believe this numerical clarification enhances the discussion of near-fault effects and further strengthens the relevance of incorporating directivity in site-specific seismic design.

26. Figure 10 clearly compares different spectra, but the legend could be improved for readability (e.g., use thicker lines for "Target Spectrum" and "TEC/ZO" to differentiate them more clearly).

**Author Response:** Thank you for the comment. We appreciate the reviewer's helpful comment regarding the visual clarity of Figure 10. As suggested, we have improved the figure by increasing the line thickness of both the Target Design Spectrum and the TEC / ZD Spectrum, while keeping the GMPE spectra in thinner, dashed, or dotted styles to maintain visual hierarchy.

These changes were made to enhance the readability and distinction of the key design spectra relative to the reference GMPE curves. The legend and color scheme were also revised for consistency and clarity.

27. The authors selected 11 records (Table 5), but the scaling procedure could be described more precisely. Please clarify: Whether spectral matching or simple scale factor was used. Whether each record meets compatibility criteria (e.g., ASCE 7–22 or Eurocode 8).

**Author Response:** Thank you for the comment. In the revised manuscript (Section 4.2), we now clarify that:

- The 11 selected ground motion records were spectrally matched to the target design spectrum using SeismoMatch (Seismosoft, 2022).
- The matching was applied over the period range 0.1–2.0 seconds, ensuring close agreement with the target across periods relevant to the dominant structural response.
- The mean spectrum of the matched records falls within ±10% of the target spectrum in this range, thus meeting the compatibility criteria outlined in Eurocode 8 (EN 1998-1:2004, Annex A).

This clarification has been incorporated into the revised manuscript (lines 363-367).

28. Figures 11 and 12 present the scaled records for different sites. However, it would strengthen the section to include statistical metrics (e.g., mean ± σ, envelope exceedance rate, number of records exceeding the target at critical periods).

**Author Response:** Thank you for the comment. In response, we have significantly enhanced the analysis and presentation of the scaled ground motion records to incorporate the requested statistical metrics:

1. Mean ± σ Envelope: For each station, the spectral acceleration (Sae) values of the scaled records were statistically evaluated, and the mean spectrum with its ±1 standard deviation (σ) envelope was plotted. This provides a clear depiction of the central tendency and variability of the selected motions (see revised Figures 11 and 12 (now labeled as Figure 13)).
2. Median Spectrum: We added the median spectrum to each figure as a robust central estimate, aligning with standard practices in probabilistic seismic hazard and site response studies (e.g., PEER-GMSM Guidelines, 2011).
3. Envelope Exceedance Statistics: A new table (Table 6) was introduced, reporting for each spectral period:
   - the mean Sae,
   - standard deviation,
   - number of records exceeding the target spectrum, and
   - exceedance rate (% of total).
4. Graphical Representations: Additionally, two new supporting figures were added:
   - One showing the mean ± σ band together with individual records and the target spectrum, and
   - Another showing the exceedance rate per spectral period, offering a quantitative measure of how well the record set envelopes the target spectrum.

All modifications have been clearly referenced and discussed in the revised manuscript (see Section 4.1 and Figures 11–13, Table 2).

29. Figure 13 effectively compares median responses to TEC and revised spectrum. However, consider labeling the TEC soil class boundaries (ZD, ZE) more explicitly in the figure or caption for clearer comparison.

**Author Response:** Thank you for the comment. In the revised version of Figure 13, the spectral curves corresponding to the Turkish Earthquake Code (TEC, 2018) Site Classes ZE and ZD have been clearly labeled as **"TEC / ZE SITE CLASS"** and **"TEC / ZD SITE CLASS"**, respectively. Additionally, the figure caption has been updated to explicitly reference these classifications to avoid any ambiguity during interpretation.

30. Table 6 shows valuable correction factors (F1/TEC vs. F1/SAR), but the implications should be briefly discussed: e.g., how these high ratios (e.g., 4.76 at Bornova) indicate potential underestimation of seismic demand in current TEC-based designs.

**Author Response:** Thank you for the comment. In the revised manuscript, we have added a brief but focused discussion on the implications of the correction factors (F1_SAR / F1_TEC) presented in Table 7.

Specifically, the observed ratios—such as 4.76 at Bornova, 3.84 at Bayraklı, and 2.93 at Karşıyaka—highlight that the current TEC-based site modification factors (F1) significantly underestimate long-period spectral demands, particularly in deep alluvial basins and soft-soil conditions. These findings suggest that relying solely on soil classification-based factors (ZE, ZD) may lead to non-conservative seismic design in such regions. The discrepancy originates from the inability of code-defined factors to capture basin-induced effects, resonance behavior, and nonlinear amplification that are revealed through site-specific response analyses.

This interpretation is consistent with recent literature emphasizing the limitations of generic code-based spectra in complex geological settings (Stewart, J. P., & Seyhan, E. (2013); Pitilakis et al., 2013). The discussion has been integrated into Section [4.2] of the revised manuscript, following Table 7.

Section 5: Summary and Conclusions

31. The section successfully summarizes the methodology and major findings. However, the conclusions would benefit from more explicit quantitative insights (e.g., amplification ratios, return periods, percent deviations) rather than mostly qualitative descriptions.

**Author Response:** Thank you for the comment. In the revised version of Section 5 – Summary and Conclusions, we have incorporated several explicit quantitative findings to strengthen the scientific clarity of the conclusions. These additions include:

- Amplification Ratios: Median surface spectral accelerations were found to be amplified by factors ranging from 2.5 to 4.8 for periods around $T = 1.0$ s, depending on the site.
- Design Parameter Comparison: The $F_1$ factors defined in TEC (2018) were compared with those derived from site-specific response analysis (SAR), and a detailed summary was presented in Table 6. For instance, in Bornova, the calculated $F_1\_SAR / F_1\_TEC$ ratio reached 4.76, indicating significant underestimation of seismic demand using code-based provisions.
- Percent Deviations from TEC: The developed scenario-based target spectrum, which incorporates near-field and directivity effects, exceeds the standard TEC spectrum by 65–80% in spectral amplitude in the 0.3–1.0 s period range.
- Return Period Specification: The selected target spectrum corresponds to a 475-year return period (10% probability of exceedance in 50 years), in accordance with TEC guidelines and consistent with probabilistic seismic hazard levels.

These quantitative results have been integrated directly into the conclusion section to support the key findings and clarify their engineering relevance.

32. The statement that "TEC factors should be corrected by at least 2.50 times" (lines 310–312) is significant. I recommend the authors provide a brief justification or numerical basis here (e.g., comparison table or reference to Figure 13 or Table 6).

**Author Response:** Thank you for the comment.The statement regarding the need to correct TEC-based modification factors is now supported with explicit numerical comparisons in the text (lines 487–497), including site-specific median Sae values at T = 1.0 s and their ratios to the TEC-defined values. These ratios (ranging from 2.93 to 4.76) clearly indicate significant underestimation of seismic demand, particularly for deep alluvial sites.

33. The authors mention "buildings between 3–10 stories" (line 313), which is important in practice. However, linking this to code performance levels (e.g., life safety, collapse prevention) would enhance the engineering relevance.

**Author Response:** Thank you for the comment. In the revised manuscript, we expanded the discussion to briefly emphasize that the observed spectral amplifications in the 0.5–1.5 s period range critically affect mid-rise buildings (3–10 stories), which correspond to this period band. A new sentence was added to highlight how this condition may influence the satisfaction of expected performance levels (e.g., Life Safety, Collapse Prevention) in deep alluvial basins. This improvement reinforces the engineering significance of our findings while maintaining conciseness in the conclusion section.

34. The last sentence (lines 315–316) mentions that future studies will explore basin effects. While this is appreciated, I suggest specifying how the presented study lays the groundwork for this (e.g., validation of deep soil profiles, limitations of 1D analysis).

**Author Response:** Thank you for the comment. In response, we have clarified in the revised manuscript that the present study forms a foundation for future basin-scale investigations through the use of calibrated 1D models, developed based on well-characterized deep soil profiles across three representative regions. As acknowledged in the limitations section, 1D equivalent-linear models cannot fully capture complex wave propagation phenomena such as edge-generated surface waves or 3D basin effects. The consistent site-dependent amplification patterns observed in this study highlight the importance of advancing toward multi-dimensional analyses in future research.

35. Overall, this section would benefit from a clearer separation between key conclusions and recommendations. Consider formatting them as two separate bullet lists (e.g., "Main Findings" vs. "Recommendations for Future Work").

**Author Response:** Thank you for the comment. Accordingly, the revised manuscript now separates the content into two distinct bullet-point lists titled "Main Findings" and "Recommendations for Future Work". This structural revision enhances readability and ensures that the study's key results and forward-looking perspectives are more clearly distinguished.

Final Recommendation: Minor Revision

The manuscript presents a valuable, technically sound, and well-structured contribution to seismic hazard evaluation and future earthquake simulation in İzmir. With minor improvements in figure clarity, language quality, and justification of novelty, the paper will be ready for publication.

**Author Response:** Thank you for the comment. We sincerely thank the reviewers and the handling editor for their valuable feedback, which significantly improved the clarity and quality of our manuscript. All minor revisions regarding figure presentation, language polishing, and justification of the study's novelty have been carefully addressed in the revised version. We hope the manuscript now meets the expectations for publication.

---

## Author Comment (AC2)

Response to Reviewer Comments

Manuscript ID: EGUsphere-2024-3488

Title: Critical Evaluation of Strong Ground Motions in Izmir and Implications for Future Earthquake Simulation Results

Dear Editor and Reviewers.

We sincerely thank the Editor and both Reviewers for their thorough evaluation of our manuscript and for the constructive comments that greatly contributed to improving the quality, clarity, and scientific rigor of the study. We carefully reviewed each comment and revised the manuscript accordingly. Major methodological explanations have been expanded, figures and tables have been upgraded for resolution and readability, and several sections—particularly Sections 3 and 4—have been completely rewritten to ensure transparency, reproducibility, and coherence of the analytical framework.

Reviewer comments are shown in blue, while our responses are shown in black. All corresponding changes have been incorporated into the revised manuscript, with clear improvements in methodological description, figure consistency, and overall narrative flow. In the revised manuscript, all changes made in response to reviewer comments are highlighted in yellow for ease of reference.

We are grateful for the reviewers' insightful contributions and believe that the revised manuscript is now significantly strengthened as a result.

Respectfully,

Assist. Prof. Dr. Şahin Çağlar Tuna

Corresponding Author, Yaşar University

This manuscript tries to evaluate ground motion characteristics and potential ground motions by future earthquake in Izmir city, Turkey. As Turkey is prone to large earthquakes, these evaluations are considered very important. Overall procedure seems probably reasonable, but as is already pointed out by Referee 1 the manuscript needs more explanation/indication and improvement of figure quality. The comments from Referee 1 are right on point, and particularly, I strongly agree with comments 1, 2, 4, 7, 10, 12, 13, 14, 20, 25, 27, 31. In addition, I would like to indicate four major comments. Therefore, I judge this manuscript major revision.

Author Response: We sincerely thank the reviewer for this valuable overall assessment. In line with the reviewer's observations, we conducted a comprehensive revision of the manuscript focusing on both scientific clarity and presentation quality. All comments highlighted by the reviewer—including those specifically emphasized (1, 2, 4, 7, 10, 12, 13, 14, 20, 25, 27, 31)—as well as the full set of comments from Reviewer 1 have been addressed in detail. Substantial improvements were made to strengthen methodological explanations, clarify assumptions, and enhance the consistency of the narrative. In addition, all figures and tables were revised for higher resolution, improved readability, and clearer scientific interpretation. These modifications significantly improve the manuscript's structure, transparency, and technical rigor. We appreciate the reviewer's guidance and believe the revised version satisfactorily resolves the concerns raised.

**Major comments:**

1. In Section 2, the author seeks for the most appropriate GMPEs for Izmir dataset considering PGA data. However, in Section 4, response spectra are evaluated using the selected GMPEs. Why don't the author evaluate the appropriateness of GMPEs using response spectra? It is necessary to show the validity of use of PGA for the evaluation of GMPEs.

**Author Response:** Thank you for the comment. We agree that the connection between the PGA-based GMPE evaluation in Section 2 and the spectral analyses in Section 4 must be clearly explained.

Our use of PGA as the primary IM for evaluating GMPE performance follows standard practice in GMPE selection literature, where PGA (or very-short-period Sa) is commonly employed as the first-order discriminator for identifying suitable ground-motion models.

This approach is consistently adopted in NGA-West1/2, Kalkan & Gülkan (2004), Boore et al. (2014), Campbell & Bozorgnia (2014), Karaca et al. (2021), and Huang et al. (2023). The underlying reasons are well established:

- PGA exhibits the lowest model-to-model epistemic variability, providing a stable basis for ranking GMPEs.
- PGA is a directly recorded quantity, whereas spectral ordinates require additional processing steps (baseline correction, filtering, SDOF transformation).
- Short-period spectral accelerations ( $T \le 0.2$  s) are strongly correlated with PGA, a relationship explicitly embedded in many GMPE functional forms.
- GMPEs are best constrained and show the lowest aleatory variability at short periods, while long-period Sa(T) inherently displays much larger dispersion.

Therefore, the use of PGA provides a statistically robust, noise-insensitive, and internationally recognized basis for GMPE screening. Our methodological approach follows this established practice.

Recent studies that adopt a similar two-stage workflow (PGA-based GMPE screening followed by physics-based spectral construction) generally do not perform a full Sa(T)–GMPE residual analysis before defining scenario spectra. Instead, GMPEs are commonly used to establish the median short-period level on rock, while the longer-period spectral shape is adjusted using additional physical considerations such as directivity, basin response, and site-specific amplification. Examples include Huang et al. (2023), Wang et al. (2022), Karaca et al. (2021), and Cetin & Çakır (2023). Although these studies differ in details, their general methodological structure aligns with the approach adopted here.

In Section 4, the selected GMPEs (CB14 and BSSA14) are not used as full spectral predictors. They serve only to:

- define the rock-level hazard (PGA and very-short-period Sa) for the Mw 6.5 deterministic scenario, and
- anchor the short-period portion of the target spectrum.

The long-period portion of the spectrum (1–2 s) is then modified using near-fault directivity and basin-amplification effects, constrained by empirical observations from the 2020 Samos earthquake (Somerville et al., 1997; Shahi & Baker, 2011; Cetin et al., 2022; Cetin et al., 2023).

Thus, Sa(T) in Section 4 is not a GMPE output but a GMPE-anchored, physics-based spectrum that reflects validated site-response behavior.

Finally, Figure 10 provides a meaningful spectral-level consistency check:

- CB14 and BSSA14 median trends align well with the target spectrum at short periods (where GMPEs are most reliable),
- while deviations at longer periods reflect intended physical modifications, not a GMPE inconsistency.

These modifications result from near-fault directivity, basin amplification, and nonlinear SRA behavior validated in Section 3. Therefore, the PGA-based GMPE selection in Section 2 is methodologically consistent with, and fully supportive of, the spectral analyses performed in Section 4.

2. It is difficult to understand how the author actually evaluates the amplification of three stations in Izmir in Section 3, because the description on the method lacks details, the resolution of figures is very low, and the figures do not have appropriate legends and/or captions. As far as I guess from the manuscript, the author obtains the SRA results by inputting ground motion recorded at 3514 site to underground structure model shown in Table 4 and Figure 5. However, underground structure models of 3513, 3522, and 3519 sites shown in Table 4 and Figure 5 do not reach bedrock,

and therefore, I am wondering at which depth the bedrock ground motion is input, which very much affects site amplification.

**Author Response:** Thank you for the comment. We agree that the original text in Section 3 did not provide sufficient detail regarding how the input rock motion was applied and how the soil columns were defined down to the engineering bedrock. These clarifications have now been fully incorporated into the revised manuscript.

Although Table 4 and Figure 6 present only the upper portions of the soil profiles for readability, each 1D site-response model was constructed using the full geotechnical and geophysical logs available for the three stations. In DeepSoil, the soil columns were extended down to the engineering bedrock, defined as the depth where  $Vs \approx 760$  m/s, consistent with standard SRA practice. The resulting total model depths are:

Station 3513: ~120 m
Station 3519: ~250 m
Station 3522: ~90 m

The 3514 rock-outcrop motion, which corresponds to the only rock site in the study area ( $Vs_{30} = 836 \text{ m/s}$ ), was applied as the within-motion (rock-outcrop boundary) at these bedrock depths in all simulations. This is now explicitly stated in the revised Section 3 to eliminate ambiguity.

To address the reviewer's concern about figure clarity, all figures related to the SRA validation (previous Figures 7–9) have been regenerated at high resolution, legends and axes have been standardized, and captions have been expanded to clearly explain the comparison between recorded and simulated spectra. The previous Figure 5 has been reorganized and updated as the new Figure 6 to more clearly show the subsurface profiles and station layout.

3. The detailed analytic procedure of Section 4 is also vague. Similar to Section 3, the manuscript just names software for analysis and lacks detailed methodology.

**Author Response:** Thank you for the comment. In response, Section 4 has been completely rewritten, expanded, and reorganized to present a transparent, step-by-step, and fully reproducible procedure. The major improvements are summarized below:

- 1. Full methodological rewrite of Section 4: The entire section was rewritten to clearly describe how the deterministic scenario, target spectrum, ground-motion selection, spectral matching, and nonlinear response analyses were conducted. All conceptual gaps in the original version have been removed.
- 2. Clarified justification for the Mw 6.5 scenario: We added a scientific explanation showing that the RADIUS (1997) scenario is consistent with modern tectonic models (EFSM20, ESHM20), confirming that Mw 6.5–6.7 remains a realistic magnitude capability for the İzmir Fault.
- 3. Rupture directivity and long-period amplification integrated analytically: The revised text now explains:

- the physical mechanism of directivity,
- why it affects 1–2 s periods,
- and how it was identified in the 2020 Samos recordings  $(1.6-2.1 \times amplification at Station 3519)$ .

These effects are now explicitly incorporated into target spectrum construction.

- 4. Step-by-step description of target spectrum development: The manuscript now clearly describes that:
  - short-period amplitudes are anchored using CB14 and BSSA14,
  - long-period ordinates are shaped using directivity, basin effects, and validated nonlinear site-response results (Section 3),
  - the final spectrum envelopes both GMPEs and TEC-2018.

This workflow is now presented as a reproducible analytical process, not just a conceptual description.

- 5. Ground-motion selection and spectral matching fully detailed: The previous brief reference to SeismoMatch has been replaced with:
  - justification of the 0.1–2.0 s matching range,
  - Eurocode 8 and PEER-GMSM compatibility requirements,
  - explanation of how mean  $\pm 1\sigma$  and exceedance rates were evaluated,
  - a clear description of how 11 NGA-West2 motions were selected.
- 6. Explicit link to validated SRA models from Section 3: We clarified that the scenario analyses used the same calibrated DeepSoil models validated with the Samos earthquake. Additional explanations were added regarding:
  - median surface spectra,
  - $\pm 1\sigma$  dispersion,
  - frequency-dependent amplification at each site.
- 7. Comprehensive reinterpretation of TEC-based amplification factors: The revised section now provides a clear engineering comparison between:
  - simulation-based Site Amplification Ratios (SAR), and
  - TEC-2018 factors (F1 and FS), highlighting long-period underestimation and structural implications.
- 8. Figures fully replaced and upgraded: All figures related to Section 4 (Figures 10–14) were regenerated with:
  - high resolution,
  - proper legends,

- improved axes,
- detailed captions.
- 9. Reorganized structure for clarity (Sections 4.1, 4.2, 4.3): The analytical workflow is now clearly separated into:
  - target spectrum formulation,
  - ground-motion selection and matching,
  - nonlinear response results and engineering implications.

These revisions directly resolve the reviewer's concern. Section 4 now presents a complete, rigorous, and reproducible analytical methodology, matching the level of detail expected in performance-based seismic hazard and site-response studies.

4. Appropriate indication and acknowledgements to data resources need to be added. Particularly, data listed in Table 5 includes data from several observation networks or institutes. Acknowledgements to continuous efforts to obtain data support such networks and institutes.

**Author Response:** Thank you for the comment. We fully agree, and the revised manuscript now includes explicit and detailed attribution to all data sources.

This study utilized two primary datasets:

- 1. AFAD (Disaster and Emergency Management Authority of Türkiye) provider of all 2020 Samos Earthquake recordings and the station data used in the site-response validation analyses in Section 3.
- 2. PEER NGA-West2 Ground Motion Database provider of the 11 real ground-motion records listed in Table 5, which were used for spectral matching and deterministic scenario analyses in Section 4.

To ensure full transparency and proper credit, we have incorporated the following revisions:

In Section 3, the description of the site-response validation has been rewritten to reflect the updated formulation, where we now state that the 3514 rock-outcrop motion used in the SRA validation was obtained from the AFAD strong-motion network, and that the 2020 Samos Earthquake recordings from AFAD formed the basis of the calibration of the site-response models. This updated text provides clearer context on how and where the reference motion was obtained, fully replacing the earlier brief wording.

In Section 4.1, we specify that all ground-motion records used for spectral matching and scenario simulations were obtained from the PEER NGA-West2 Ground Motion Database.

Finally, the Acknowledgements section has been revised to include the following explicit acknowledgement: "The authors gratefully acknowledge AFAD for providing the strong-motion recordings of the 2020 Samos Earthquake, and the Pacific Earthquake Engineering Research Center (PEER) for maintaining and providing access to the NGA-West2 ground-motion database.

The continuous efforts of these observation networks and contributing institutions are sincerely appreciated."

These revisions ensure that all data providers are appropriately credited and that the updated Section 3 text remains fully consistent with the reviewer response.

Minor comments: - Table 2: Style of fonts needs to be consistent in the table.

**Author Response:** Thank you for the comment. The table has now been fully revised to ensure consistent font style, size, and formatting throughout.

- p.8, line 166: "The city covers large areas of alluvial soil condition" seems grammatically incorrect. "The city is covered with large areas of alluvial soil condition" may be better.

**Author Response:** Thank you for the comment. The original sentence has been fully revised for clarity and grammatical correctness. The entire paragraph describing the 2020 Samos Earthquake and the selection of validation sites has been rewritten. The revised text no longer contains the incorrect sentence and now reads (lines 229-234):

"The earthquake occurred on October 30, 2020, with a moment magnitude of Mw = 6.9 and an epicenter located approximately 14 km northeast of the Greek island of Samos, within the Aegean Sea. Despite the offshore origin, the event caused significant structural damage in İzmir, largely attributed to shallow rupture depth and site amplification effects. Numerous strong motion stations across İzmir captured the event (Figure 5), including sites situated on soft alluvial soils and others on rock outcrops, enabling a comparative evaluation of site effects (Kwok et al., 2007; Kramer, 1996)."

- p.9, line 174: "data's" seems grammatically incorrect. "data" may be better.

**Author Response:** Thank you for the comment. The sentence has been fully revised, and the incorrect phrase "PGA data's" has been removed. The entire methodological description in this section has been rewritten for clarity and scientific accuracy. The revised text now reads (lines 235-245):

"The 1D SRA simulations were conducted by propagating the 3514 rock-outcrop motion—obtained from the AFAD strong-motion network—through the calibrated soil columns at the selected sites. Station 3514, characterized by a Vs30 value of 836 m/s, is the only rock site in the study area and therefore provides an appropriate outcrop reference motion for validation purposes.

Four representative locations were selected to capture the geotechnical and geological variability across İzmir:

- Karşıyaka (3519) basin edge, thick alluvial deposits
- Bayraklı (3513) deep soft soils with high amplification potential
- Bornova (3522) moderately deep alluvial layers
- Konak (central district) urban area with transitional soil conditions

These stations collectively reflect the diversity of soil conditions and shaking characteristics across İzmir, enabling the development of a calibrated and reliable SRA model for subsequent scenario-based simulations."

- Figure 6: It is better to set the same size for top and bottom panels.

**Author Response:** Thank you for the comment. In the revised manuscript, the original multi-panel Figure 6—showing generic modulus-reduction and damping-ratio curves—has been removed. These curves represent well-established literature models (Seed & Idriss, 1970; Vucetic & Dobry, 1991), and including them was not essential to the presentation of the study's results.

The methodology now cites these references directly in the text:

"Modulus reduction  $(G/G_{max})$  and damping ratio (D) curves were assigned based on soil classification. The Seed and Idriss (1970) model was used for cohesionless soils, while the Vucetic and Dobry (1991) curves were adopted for cohesive soils."

- Table 5: Style of the table needs improvements (distorted aspect ratio of fonts, inconsistent for centering or right-aligning, etc).

**Author Response:** Thank you for the comment. Table 5 has been completely reformatted in the revised manuscript. All fonts have been standardized, column alignment has been corrected (text centered, numerical fields right-aligned), and row spacing has been normalized. The updated version is now included as Table 6 in the revised manuscript.